# Quantifying Functional Impairment of *ABCA3* Variants Associated with Interstitial Lung Disease

**DOI:** 10.3390/ijms24087554

**Published:** 2023-04-20

**Authors:** Xiaohua Yang, Christina K. Rapp, Yang Li, Maria Forstner, Matthias Griese

**Affiliations:** 1Dr. von Haunersches Kinderspital, German Center for Lung Research (DZL), University of Munich, Lindwurmstr. 4a, D-80337 Munich, Germany; xiaohua.yang@med.uni-muenchen.de (X.Y.); maria_elisabeth.forstner@med.uni-muenchen.de (M.F.); 2Medical College, Chongqing University, Chongqing 400030, China

**Keywords:** ATP-binding cassette subfamily A member 3, ABCA3, trafficking, pumping, function, subgroups, phosphatidylcholine, interstitial lung disease

## Abstract

ATP-binding cassette subfamily A member 3 (ABCA3) is a lipid transporter within alveolar type II cells. Patients with bi-allelic variants in *ABCA3* may suffer from a variable severity of interstitial lung disease. We characterized and quantified ABCA3 variants’ overall lipid transport function by assessing the in vitro impairment of its intracellular trafficking and pumping activity. We expressed the results relative to the wild type, integrated the quantitative readouts from eight different assays and used newly generated data combined with previous results to correlate the variants’ function and clinical phenotype. We differentiated normal (within 1 normalized standard deviation (nSD) of the wild-type mean), impaired (within 1 to 3 nSD) and defective (beyond 3 nSD) variants. The transport of phosphatidylcholine from the recycling pathway into ABCA3^+^ vesicles proved sensitive to the variants’ dysfunction. The sum of the quantitated trafficking and pumping predicted a clinical outcome. More than an approximately 50% loss of function was associated with considerable morbidity and mortality. The in vitro quantification of ABCA3 function enables detailed variant characterization, substantially improves the phenotype prediction of genetic variants and possibly supports future treatment decisions.

## 1. Introduction

Pathogenic variants in the gene for lipid transporter ATP-binding cassette subfamily A member 3 (ABCA3) are the most prevalent known monogenic cause for neonatal surfactant dysfunction syndrome and interstitial lung disease in children (chILD) and young adults [1]. Depending on the degree of ABCA3 deficiency, a broad range of diffuse parenchymal lung diseases ranging from respiratory failure in mature newborns and early death, chronic interstitial lung disease with the progressive development of lung fibrosis, and little or no respiratory disease at all, may evolve [2,3].

Pulmonary surfactant reduces the alveolar surface tension, is produced in alveolar type II cells and stored in lamellar bodies (LBs) [4]. ABCA3 is involved in LB biogenesis and lipid transport into LBs [5]. After translation of the *ABCA3* gene, ABCA3 moves from the endoplasmic reticulum (ER) to the Golgi apparatus, where N-glycosylation takes place (Figure 1). Then, ABCA3 is routed via multivesicular bodies (MVBs) to the outer membrane of LBs [6,7,8]. When the surfactant–lipoprotein complex is released from the LB via regulated exocytosis, ABCA3 can be recycled for a new generation of LBs or will be degraded in lysosomes [7,9]. 

Initially, the genomic information of the *ABCA3* variants was used to categorize the outcome of ABCA3-deficient patients [10]. Nonsense and frameshift variants, also known as loss-of-function variants, commonly led to a “null” phenotype, with neonatal respiratory distress syndrome and death within the first months of life, whereas “other” (i.e., missense, in-frame insertions and deletions (indels), or splicing variants) might be compatible with a milder phenotype and survival [10]. Then, in vitro studies significantly advanced cell biological knowledge on the functional performance of different variants [5,6,8,11,12,13,14,15,16,17,18,19,20,21,22,23]. Currently, two major types of *ABCA3* variants are distinguished: trafficking mutations causing abnormal intracellular localization and leading to complete lipid transport deficiency and mutations that do not affect the trafficking and correct localization of the ABCA3 protein but impair its lipid transport [11,13,14,20,23]. Currently, we lack an approach to quantitatively capture and combine all readouts from different assays so that correlations among data from the cellular level, from the genomic prediction and from the clinical outcome of patients with various *ABCA3* variants can be determined [2,10,11,14,19]. 

Here, we (1) systematically explore ABCA3 functional assays and suggest a quantification of the cellular readouts suitable for comparison between studies, (2) we demonstrate the benefit of PC recycling in comparison to de novo PC synthesis as a functional variable, (3) we generate a significant body of new data on not yet investigated *ABCA3* variants and (4) we correlate these in vitro results to patients’ clinical course.

## 2. Results

For reference, the results in the WT ABCA3 transfected cells without treatment were used, and the normal ranges were calculated for each variable (Table 1, column 3). For comparison with the ABCA3 variants, the absolute values were normalized to one, and the corresponding nSDs were calculated. Then, their mean, median and range were calculated (Table 1, columns 4 to 7).

In addition to the variants previously assessed by our group, we selected and expressed 10 novel *ABCA3* variants (F1077I, P248S, D953H, G202R, P32S, E1364K, C611R, Q1045R, G1314E and G571R) and 4 known *ABCA3* variants (E292V, A1046E, V1399M and G1421R) from which we had patients´ clinical data. We quantified their intracellular trafficking and pumping characteristics in vitro (Appendix A). These ABCA3 variants had variable deviations from normal ABCA3 function, and the results of this large number of experiments are described in the following. 

### 2.1. Intracellular Trafficking of ABCA3 Variants

The different variants exhibited a broad range of % ER localizations (Figure 2A). The majority of the variants assessed was within a 1 + 1 nSD range of WT ER localization, indicating no significant abnormality. Some variants, such as F1203del, R280C, N124Q and N140Q, had increased retention in the ER and were consequently classified as “impaired”. The variants L982P, M760R, Q215K and G571R did not even reach the ER, indicating defective ER localization. On the other hand, the variants L101P, G1518fs, L1553P and Q1591P were enriched in the ER, with values above 1 + 3 nSD (up to 5-fold above WT), indicating abnormal ER retention and, thus, also classified as “defective” (Figure 2A). 

As expected, the variants with abnormal ER localization resulted in abnormal downstream trafficking. The variants that were retained in the ER (L101P, G1518fs, L1553P and Q1591P) or did not even enter the ER (L982P, M760R and Q215K) were completely defective in N-glycosylation, lysosomal compartment localization and proteolytic cleavage (Figure 2B–D). If the N-glycosylation of an ABCA3 variant was abnormal, the ratio of complex oligosaccharide protein to hybrid oligosaccharide protein was reduced. Some variants with impaired N-glycosylation, such as N568D, L1580P and G1221S, could still be processed to lysosomal compartments and then underwent proteolytic cleavage (Figure 2B). The ABCA3 variants resulted in diverse aberrant lysosomal localizations. Few had impaired localization, and many had defective localization patterns (Figure 2C). Several variants with abnormal lysosomal localization had a normal level of proteolytic cleavage (D953H, C611R and P32S) and others, such as K1388N and S1262G, had impaired proteolytic cleavage, even with normal upstream processing (Figure 2D). Based on these data on ABCA3 intracellular trafficking, we differentiated the ABCA3 variants with defective, impaired and normal trafficking and sorted the variants accordingly (Figure 2 and Appendix A). 

### 2.2. Differentiation of PC De Novo Synthesis Pathway and PC Recycling Pathway 

To better understand the involvement of the ABCA3 protein in the intracellular PC lipid metabolic pathway, we simultaneously used two different labels for PC [16,17] (Figure 3A,B and Appendix A). Propargyl-choline was incorporated into the ER after 4 h and into the lysosomal compartment after 8 h. The TopF-PC never colocalized with the ER marker, but it started to mark the lysosomal compartment already after 4 h (Figure 3A,B). We then assessed the time-dependent characteristics of the ABCA3^+^ lipid filling as a function of the label used (Figure 3C–F). The incorporation of propargyl-choline into ABCA3^+^ vesicles increased within 48 h to a maximum of 100% (Figure 3E). In contrast, TopF-PC in ABCA3^+^ vesicles reached a maximum within 12 h but at a clearly lower level of approximately 90% and then decreased again to approximately 70% at 24 h and 48 h (Figure 3C–E). These results were confirmed when we incubated ABCA3-HA WT cells with 100 μM propargyl-choline and with 1:20 TopF-PC in parallel but separately. 

### 2.3. Pumping Dysfunction of ABCA3 Variants In Vitro

The variants with defective trafficking never had any activity in all pumping assays (L101P, L982P, M760R, Q215K, L1553P, Q1591P, G571R and G1518fs; Figure 4, dark-grey shadow). 

All ABCA3 variants defined as having impaired trafficking had underdeveloped ABCA3^+^ vesicle sizes and a defective PC metabolism from recycling (Figure 4, light-grey shadow). Generally, the PC metabolism from recycling was much more disturbed than the PC metabolism from de novo synthesis. Several variants that had a defective PC metabolism from recycling still had a normal or slightly impaired PC from de novo synthesis (K1388N, N568D, L1580P and A1046E) (Figure 4B–C).

Among the variants with normal trafficking but impaired pumping, those with normal ABCA3^+^ vesicle size also had normal PC metabolism results (Figure 4, no shadow; Appendix A). Of note, R288K was the only identified variant with a normal PC metabolism in both de novo synthesis and from recycling, despite deficient ATPase activity. All ABCA3 variants that had data on ATPase activity showed a decreased readout, except T1114S with normal ATPase activity and E690K with significantly higher ATPase activity than the WT. 

By integrating results from trafficking and pumping assays of ABCA3 variants, we differentiated three groups of ABCA3 variants in vitro: normal trafficking but impaired pumping, impaired trafficking and impaired pumping, and defective trafficking with defective pumping variants (Appendix A).

### 2.4. Clinical Phenotype and Function of ABCA3 Variants

To assess the correlation between patients’ outcome and ABCA3 function, we first lined up the individual ABCA3 variants (sum of the trafficking and pumping activities (Figure 5A)) and the contribution of the trafficking and the pumping activities (Figure 5B). The subjects carrying variants in a homozygous constellation were aligned, as the interpretation of this is the most straight forward (Figure 5C). Early death (i.e., clinical outcome score: 5) occurred in patients with an overall in vitro function of approximately 1 of 2, i.e., a loss of 50%.

We then analyzed the results of the individual patients carrying either homozygous or compound heterozygous variants, including all those with in vitro characterized ABCA3 variants (32/54) (Figure 6). Similarly, patients with variants with an overall function below ~1.5 were at high risk for early death, and none of the subjects with a function below 1 had a good outcome. We identified a good correlation between clinical outcome and functional activity (Appendix A). Compared to pumping alone, function (i.e., sum of trafficking and pumping) better predicted the patients’ clinical outcomes. Of interest, three cases did not fit the expected relation (Figure 6 and Appendix A; Appendix A), i.e., patients 4 (compound heterozygous R208W and R43H), 15 (homozygous E690K) and 26 (homozygous K1388N) had poor clinical outcomes although relatively well preserved vitro function (all > 1.5). The trafficking was normal (#4 and 15) or slightly impaired only (#26), and the pumping was moderate but not severely impaired (Figure 6; Appendix A). In addition, there were three other patients (#27, 28 and 40) who survived into adulthood, with diffuse parenchymal lung disease and moderately impaired function in vitro. The trafficking was all mildly impaired, and the pumping was more severe (Figure 6).

It was not possible to predict the clinical phenotype of the patient based exclusively on the ACMG classification of the *ABCA3* variants (Appendix A). The implementation of the results of the functional assays moderately improved the prediction (r = 0.69, Appendix A). Among the 42 different variants assessed with ACMG, 2 were likely benign, 5 of uncertain significance, and 2 likely pathogenic, whereas the others were pathogenic. Functional characterization classified all variants as pathogenic, although to different extents (Appendix A and Figure 5). 

## 3. Discussion

A large number of *ABCA3* variants may result in human lung disease. Missense variants occur without a significant hotspot region throughout the gene. The functional impact of many *ABCA3* variants is unknown or has been determined in methodologically heterogeneous studies. Here, we propose the usage of quantitative cellular readouts for in and between study results, explore new variants and validate the PC recycling assay as a useful functional tool. 

Inconsistent variant interpretation may lead to false predictions of individual patient’s clinical outcomes and related treatment decisions. In silico predictions and structure analysis are currently not strong enough to reliably predict the effects of *ABCA3* variants [24]. This is in agreement with our results, as the interpretation of the variants based on the ACMG guidelines [25] failed to correlate to clinical outcomes. 

For the in-depth characterization of the intracellular trafficking and pumping of the ABCA3 variants, we generated new data and combined them with previous results so that we assessed multiple functional aspects of as many *ABCA3* variants as available. 

First, based on a common statistical and readily usable definition of impaired and defective, we quantitatively grouped the variants into three groups: normal trafficking but impaired pumping, impaired trafficking and impaired pumping, and defective trafficking with defective pumping. Whereas for each of these groups this approach resulted in qualitatively good agreement with the clinical outcome, it was not suitable, however, for quantitative correlation analysis. 

Then, we integrated the WT-reference functional data and assessed their prediction of clinical outcome. This approach allowed for the consideration of the quantitative degree of deviation from the WT in each assay. Additionally, the data generated here can also be used for further calculations, in particular when new or additional results on the variants will be published. Of interest may be potential explanations for unexpected deviations from the overall close genotype–phenotype correlation. Two of the patients with a poor outcome not correlating to their relatively well-preserved in vitro function had lung transplantation in their first year of life. This may have been conducted because of their overall clinical prognosis, e.g., secondary damage from artificial ventilation or other complications and was possibly not based on the prediction of ABCA3 dysfunction alone. Excluding those two subjects, a functional threshold for death in the first year of life was approximately 1 of 2 (i.e., =50% of normal trafficking and pumping sum) (Figure 6A). This refers to the in vitro residual function of a variant. For transfer to the in vivo situation, different constraints must be considered. In neonates, an immediate delivery of surfactant into the alveolar space within a few minutes of birth is necessary; in children with stable interstitial lung acute, exacerbations likely differ from the requirements of surfactant transport during chronic respiratory insufficiency or those of normal subjects carrying a WT and a null variant. Thus, dense functional data on specific variants will better predict the clinical prognosis and more precisely guide treatment decisions in the future. Similarly, we can learn from the variants with normal results in all three pumping assays. For example, R288K only had an impaired doxorubicin detoxification and defective ATPase activity [18,19,26]. This variant appears more as a risk factor than as a disease-causing variant on its own; it may only be of relevance in stressful situations for the pulmonary surfactant system, e.g., premature birth, artificial ventilation or other complications. R208W and G964D also had normal results in the three pumping assays, but patients suffered from the slowly progressing chronic ILD [18,21]. This implicates that additional mechanisms may be necessary to consider and implement into the array of functional tests for *ABCA3* variants. It is clear that ABCA3-related lung disease may be additionally affected by modifier genes or environmental factors, which are not easily explored in vitro [6,15,26,27].

For a better understanding of the potential involvement of the ABCA3 protein in the intracellular PC lipid metabolic pathways, we simultaneously used two different labels of PC [16,17]. The results suggested that these two fluorescent labels were metabolized independently in vitro, adding new evidence to former studies describing those intracellular PC lipid metabolic pathways [28,29,30]. Propargyl-choline PC found in ABCA3^+^ vesicles represents PC derived from de novo synthesis, while TopF-PC indicated PC from recycling into the ABCA3^+^ vesicles [31,32,33,34]. In this study, we found that PC from recycling was much more sensitive than PC from de novo synthesis towards the effects of functionally impaired *ABCA3* variants. PC from recycling usually contributes approximately 55–75% of surfactant PC [28,29,30]; this fraction increases to 95% in the neonatal period [35,36]. This may explain why ABCA3 variants with impaired or defective PC from recycling could lead to the phenotype of neonatal distress with surfactant dysfunction but were no longer clinically relevant later in life [5,37]. 

Our study has some shortcomings. We used A549 cells as an established in vitro model of ABCA3 variant functional assessments, with its similarity to ABCA3 protein processing and colocalization as primary alveolar type II cells [8,38]. The very similar results obtained in the A549 cells or HEC293 cells when assessing the same variants support this in vitro system (see Figure 2 and Figure 4; Appendix A). The usage of primary patient-specific cells might be superior to also account for other genetic background modifiers. Such cells are barely available and are not readily labeled for *ABCA3* to determine readouts. In the future, patient-derived induced pluripotent stem cells or such cells engineered to carry *ABCA3* variants and the appropriate reporters may be used. The presented analysis was based on heterogeneous data from novel experiments, our own previous work and from data retrieved from the literature. While differences in the expression levels from the WT and variants from different clones were taken care of in our and the other original studies, the differences between cell lines (A549 and HEC293) or other experimental details, including different genomic integration sites and transient or constitutive expression, were elegantly controlled by expressing all readouts as the ratio of the result in a variant to that in the corresponding WT under the exact experimental conditions. Lastly, clinical data from prospective studies are expected to improve the precision correlation of clinical outcomes to in vitro data.

This quantitative approach extended the fine differentiation of ABCA3 variants beyond that of abnormal intracellular protein trafficking and pumping dysfunction [23]. It is obvious that overlaps and combinations of different cellular mechanisms may occur [39]. Considering the structural similarities of ABCA3 with cystic fibrosis transmembrane conductance regulator (CFTR = ABCC7), *ABCA3* variants may also cause no protein expression, e.g., from mechanisms such as unstable truncated RNA, protease destruction of misfolded ABCA3, or reduced amounts and instability of the transporter, which were not covered here [39]. With increasing number of identified and characterized *ABCA3* variants such mechanisms could easily be accommodated in the system proposed here. 

## 4. Materials and Methods

### 4.1. Cell Culture

The A549 cells stably expressing HA-tagged wild-type (WT) or mutated ABCA3 protein (ABCA3-HA) were cultured, and stable cell clones were generated, as previously described [13,22]. Briefly, A549 cells were cotransfected with pCMV(CAT)T7-SB100 and pT2/HB-puro-ABCA3-HA WT or ABCA3 variation. The selection of stable cells was started by the addition of 1 mg/mL puromycin (Thermo Fisher Scientific, Waltham, MA, USA), and a clone (of at least 5 clones) stably expressing mutant ABCA3 protein level compared to WT ABCA3 protein level was selected (number of independent experiments = 3). 

### 4.2. Protein Isolation and Western Blotting

Cells lysis, protein isolation, concentration measurement and immunoblotting with 15 µg protein per lane were performed, as previously described [13].

### 4.3. Immunofluorescence Staining and Quantification

The immunofluorescence staining and quantification were performed, as previously described [16,17]. Briefly, we quantified the percentage of ABCA3-HA-positive vesicles or dots (hereinafter referred to as ABCA3^+^ vesicles) colocalized with lysosomal compartment marker CD63 or with the ER marker calnexin. For this, the cells were fixed and permeabilized, blocked and then ABCA3-HA protein, CD63 or calnexin were probed with anti-HA, anti-CD63 or anti-calnexin antibody (abcam, Cambridge, UK). For visualization, the appropriate AlexaFluor secondary antibodies were used (life technologies, Darmstadt, Germany). The nuclei were stained by incubation with 0.1 μg/mL 4′,6-diamidin-2- phenylindol (DAPI, life technologies, Darmstadt, Germany). To track the PC from de novo synthesis and recycling pathways in vitro, A549 cells stably expressing ABCA3-HA WT were separately incubated with 100 μM propargyl-choline (for de novo synthesis exclusively), with 1:20 TopF-PC (for recycling exclusively) or with 150 μM propargyl-choline mixed with 1:5 TopF-PC (for de novo synthesis and recycling simultaneously) for different time periods (0, 4, 8, 12, 24 and 48 h) and then stained for the ER marker calnexin, lysosomal compartment marker CD63 and ABCA3-HA. Images were acquired using a ZEISS LSM 800 with ZEN 2 blue edition software. The percentage of ABCA3^+^ vesicles colocalized with CD63 or with calnexin and the volume of ABCA3^+^ vesicles were measured using the modified Fiji (Image J) “Particle_in_Cell-3D” plugin [40].

### 4.4. Overview of the Assays Used to Characterize ABCA3 Intracellular Trafficking and Pumping Activity

We used 8 different biochemical tests to characterize the ABCA3 variants. Each assay was coded by a letter (Appendix A), and its assumed intracellular site is indicated in Figure 1. 

In the assay “% ER localization of ABCA3” (A), we assessed the percentage of ABCA3 localized within ER by co-staining immunofluorescent ABCA3^+^ vesicles and the ER markers calnexin using confocal microscopy [14,18]. In the assay “N-glycosylation of ABCA3” (B), we quantified the ratio of complex oligosaccharide protein to hybrid oligosaccharide protein after endoglycosidase H digestion by immunoblotting [18,20]. In the assay “% lysosomal localization of ABCA3” (C), we assessed the percentage of ABCA3 processed to CD63^+^ intracellular vesicle membrane by co-staining immunofluorescent ABCA3^+^ vesicles and the lysosomal marker CD63 using confocal microscopy [20]. In the assay “proteolytic cleavage of ABCA3” (D), we quantified the ratio of 170 kDa proteolytically cleaved protein band to 190 kDa noncleaved protein band by immunoblotting [18]. In the assay “volume of ABCA3^+^ vesicles” (E), we measured the volume of immunofluorescent ABCA3^+^ vesicles located inside cells using confocal microscopy [11,18]. In the assay “amount of PC from recycling into ABCA3^+^ vesicles” (F), we assessed the PC (from recycling) transport activity of ABCA3^+^ vesicles by the fluorescence intensity of TopF-PC per ABCA3^+^ vesicles in all ABCA3^+^ vesicles using confocal microscopy [17]. In the assay “amount of PC from de novo synthesis into ABCA3^+^ vesicles” (G), we assessed the PC (from de novo synthesis) transport activity of ABCA3^+^ vesicles by the fluorescence intensity of propargyl-choline per ABCA3^+^ vesicles in all ABCA3^+^ vesicles using confocal microscopy [16]. In the assay “ATPase activity of ABCA3” (H), the ATP hydrolysis activity of ABCA3 was assessed by measuring the vanadate-induced nucleotide trapping and photoaffinity labeling of ABCA3-GFP with 8-azido-[α-32P] ATP or 8-azido-[α-32P]ADP [20,41,42] or by measuring the free phosphate released by cells [14,19]. 

### 4.5. Data and Statistical Analysis

Data on the intracellular trafficking and pumping of ABCA3 variants were newly generated in this study or retrieved from our own previous experiments [11,12,13,15,16,17,18,21,22,26]. In addition, we included all published data identified in a literature research via PubMed [5,6,8,19,20,38,41,42,43]. The different data sources were labeled by corresponding symbols in the figures. All variants, their cell models and sources are listed in Appendix A. 

The mean values and standard deviation (SD) values were collected from experiments of WT cells. If no numerical data were indicated in the publications, the mean and SD values were retrieved as approximate values from graphs. The SD values are calculated as the standard error of the mean (SEM) values multiplying by the square root of n (number of independent experiments). In each assay with a specific ABCA3 variant, the readout was expressed as the ratio of the result in a variant to that in the WT (variant/WT). If in the publications there was only a description such as “similar to wild type”, the value was set to “1”. If there were no detectable ABCA3^+^ vesicles, then the value was set to “0”. If there were results from different publications or studies, the ratios and mean values were calculated as described. For all assays, the results in ratios were similar to or smaller than 1, except for the assay of the % ER localization. In these experiments, pathological ER retention increased the ratio to values larger than 1. To align these data, for such values above 1 the reciprocals were used (1/value of variant).

For each type of experiment, we defined normal ranges based on the variations observed in experiments with WT ABCA3 cells in the absence of treatments. For this, the SD in a certain experiment was divided by the mean and defined as the normalized SD (nSD). Based on all nSDs of a certain biochemical test, the mean, median and range of nSDs were calculated (Table 1). The mean value of the WT was set to 1. We defined all results as “normal” within the range of 1 + 1 nSD, as “impaired” within 1 + 3 nSD (covering 99.7% of all observed WT values) and as “defective” beyond 1 + 3 nSD (Table 1; Figure 2 and Figure 4). This approach allowed for a quantitative assessment of the deviations of the different ABCA3 variants from normal. All results of the WT cells were obtained under the same conditions as the corresponding experiments with the various ABCA3 variants. 

The variants with defects in all four trafficking assays were classified as defective trafficking (Figure 2, dark-grey shadow). Variants with some impairments in at least one but not all four trafficking assays were classified as impaired trafficking (Figure 2, light-grey shadow). Variants with a normal range in all four trafficking assays were classified as normal trafficking (Figure 2, no shadow). Pumping of ABCA3 was classified following the same principle: defective if all assays showed defective pumping, impaired if at least one assay was impaired/defective, and normal if all four pumping assays were in normal range (Figure 4, Appendix A). Both variables were taken into account to characterize the function of ABCA3 variants in vitro using the following quantitative rating: 0, normal trafficking and normal pumping; 1, normal trafficking with impaired pumping; 2, normal trafficking with defective pumping; 3, impaired trafficking and impaired pumping; 4, impaired trafficking and defective pumping; 5, defective trafficking and defective pumping (Appendix A). For the quantitative analysis, the average trafficking and pumping of the ABCA3 variants (i.e., (Σ results of trafficking assays)/n, n: number of different trafficking assays, 3 ≤ n ≤ 4; (Σ results of pumping assays)/n, n: number of different pumping assays, 1 ≤ n ≤ 4) and the sum of those two variables were calculated to quantify the overall function of ABCA3. 

The interpretations of the *ABCA3* variants were based on the American College of Medical Genetics and Genomics (ACMG) guidelines [25]: “benign”, “likely benign”, “uncertain significance”, “likely pathogenic” and “pathogenic” and scored as 1, 2, 3, 4 and 5, respectively. If there was more than one variant identified on one allele of a patient, then the scoring was based on the most severe variant. 

Patients’ clinical outcome was rated as the following. Individuals alive without symptoms of any age (0–2 years in this study) were scored as 0; individuals with ILD ≥ 18 years were scored as 1; individuals with ILD < 18 years were scored as 2; individuals who died > 18 years were scored as 3; individuals who died > 1 year were scored as 4, and individuals who died < 1 year of age were scored as 5. A lung transplantation in an individual was valued equally as death from ABCA3-related lung disease. In individuals with the same identified variants but with different clinical outcomes, the average clinical outcome score was calculated and used. Subjects with only one *ABCA3* variant or variants with a lack of in vitro experiments were excluded from the correlation analysis. 

The average results are expressed as the means ± SEM. Statistical tests were performed using GraphPad Prism 7.0 (GraphPad Software, La Jolla, CA, USA). One-way ANOVA with Dunnet’s post hoc test was used to compare the mean value among multiple groups. Spearman’s analysis was used to correlate the genotype and phenotype of *ABCA3* variants. *p*-Values < 0.05 were considered statistically significant.

## 5. Conclusions

In summary, we successfully quantitated the function of many *ABCA3* variants relevant for affected patients [44]. We found overall acceptable quantitative variant–phenotype predictions and identified a threshold of an approximately 50% loss of function as being associated with considerable morbidity and mortality due to ABCA3 dysfunction. 

## Figures and Tables

**Figure 1 ijms-24-07554-f001:**
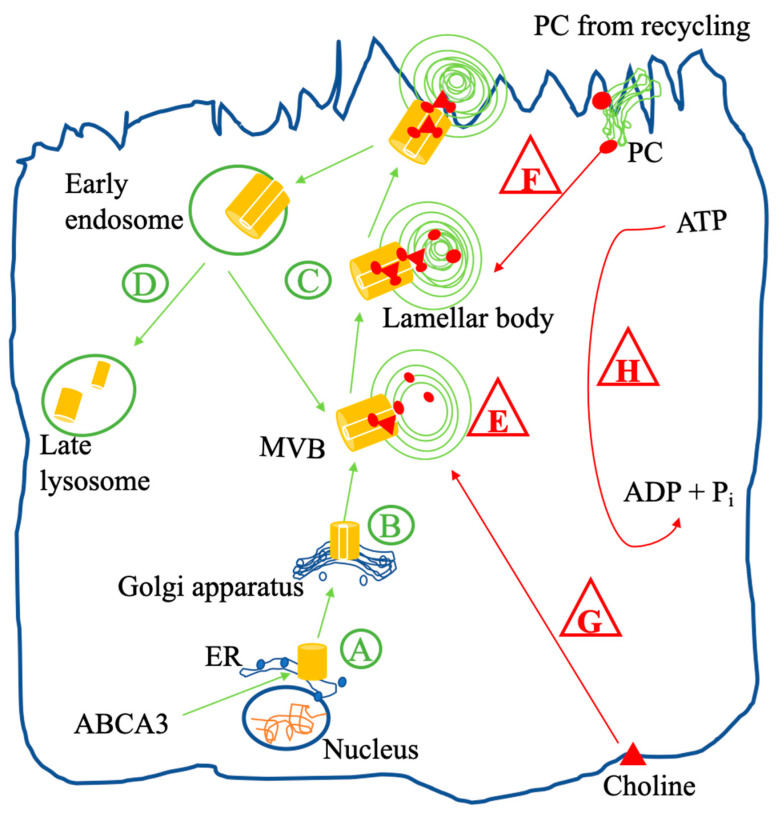
Intracellular trafficking (in green circle) and function (in red triangle) of wild-type ABCA3 in an alveolar type II cell. After synthesis, immature ABCA3 is translocated in the ER (**A**) and then trafficked to the Golgi apparatus, where N-glycosylation takes place (**B**). After N-glycosylation, ABCA3 is routed via multivesicular bodies (MVBs) to the outer membrane of lamellar bodies (LBs), which are lysosome-related compartments (**C**) involved in LB biogenesis (**E**). When the lipoprotein in LBs is released via regulated exocytosis, ABCA3 stays in the plasma membrane and then is either recycled or degraded in lysosomes, undergoing proteolytic cleavage (**D**). PC from de novo synthesis (**G**) assembles into MVBs and LBs by the ABCA3 protein with ATPase activity (**H**) and then is released as a pulmonary surfactant component via exocytosis. Surfactant DPPC is also produced by the remodeling of PC from recycled surfactant liposomes (**F**). After remodeling, DPPC is taken up by the ABCA3 protein into MVBs and LBs, which are involved in surfactant metabolism.

**Figure 2 ijms-24-07554-f002:**
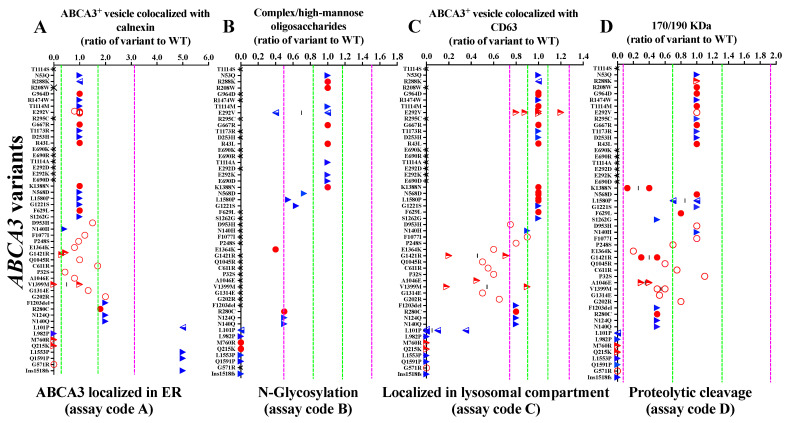
Data on the intracellular trafficking of ABCA3 variants: (**A**) localized in ER; (**B**) N-glycosylation; (**C**) localized in lysosomal compartments; (**D**) proteolytic cleavage. The green, dotted lines stand for ±1 nSD (fraction of WT nt). The purple, dotted lines stand for ±3 nSD (fraction of WT nt). The red, hollow circles represent data from this study. The red, solid circles represent data from our previous experiments. The red, semi-hollow triangles represent data from this study and the literature. The blue, semi-hollow triangles represent data from our previous experiments and the literature. The blue, solid triangles represent data from the literature. The black crosses represent that assays were not performed. Dark-grey shadow represents defective trafficking. The light-grey shadow represents impaired trafficking. No shadow represents normal trafficking. The assay code refers to Figure 1.

**Figure 3 ijms-24-07554-f003:**
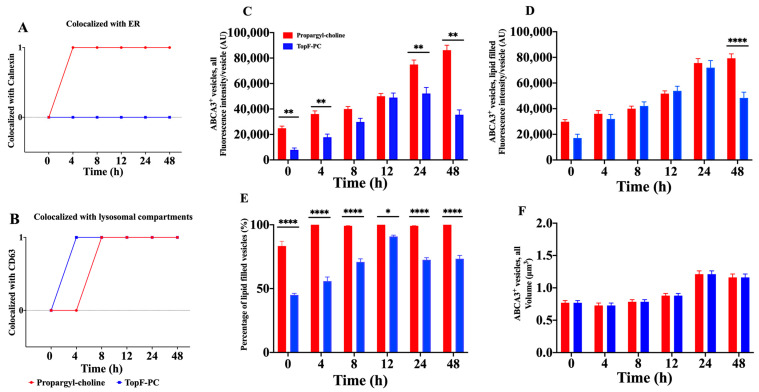
A549 cells stably expressing ABCA3-HA WT were incubated with 150 μM propargyl-choline mixed with 1:5 TopF-PC (n = 2) for different times (0 h, 4 h, 8 h, 12 h, 24 h and 48 h) and then stained for (**A**) endoplasmic reticulum (ER) marker calnexin; (**B**) lysosomal compartment marker CD63; (**C**–**F**) ABCA3-HA. Confocal images at different time points were taken (Appendix A). The results are shown as the means + SEM. One hundred and twenty ABCA3-HA-positive vesicles were randomly selected from six different fields in each condition. * *p*-Value = 0.0332, ** *p*-value = 0.0021, and **** *p*-value < 0.0001.

**Figure 4 ijms-24-07554-f004:**
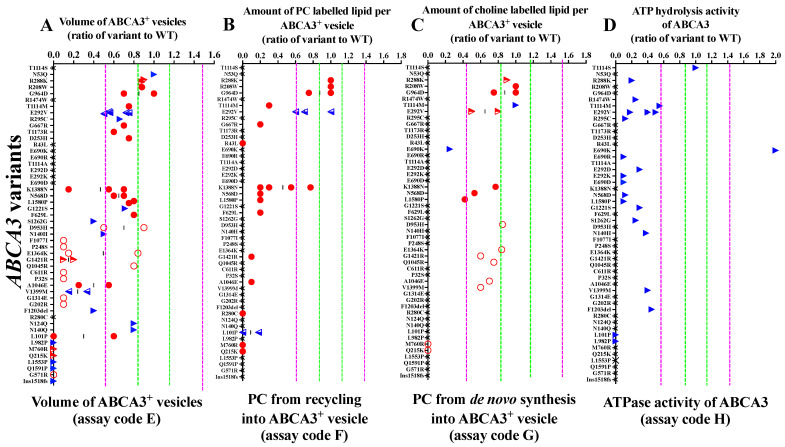
Data on the pumping of the ABCA3 variants: (**A**) volume of ABCA3^+^ vesicles; (**B**) PC from recycling into ABCA3^+^ vesicles; (**C**) PC from de novo synthesis into ABCA3^+^ vesicles; (**D**) ATPase activity of ABCA3. The green, dotted lines stand for ±1 nSD (fraction of WT nt). The purple, dotted lines stand for ±3 nSD (fraction of WT nt). The red, hollow circles represent data from this study. The red, solid circles represent data from our previous experiments. The red, semi-hollow triangles represent data from this study and the literature. The blue, semi-hollow triangles represent data from our previous experiments and the literature. The blue, solid triangles represent data from the literature. The black crosses represent that assays were not performed. The dark-grey shadow represents defective trafficking. The light-grey shadow represents impaired trafficking. No shadow: normal trafficking. The assay code refers to Figure 1.

**Figure 5 ijms-24-07554-f005:**
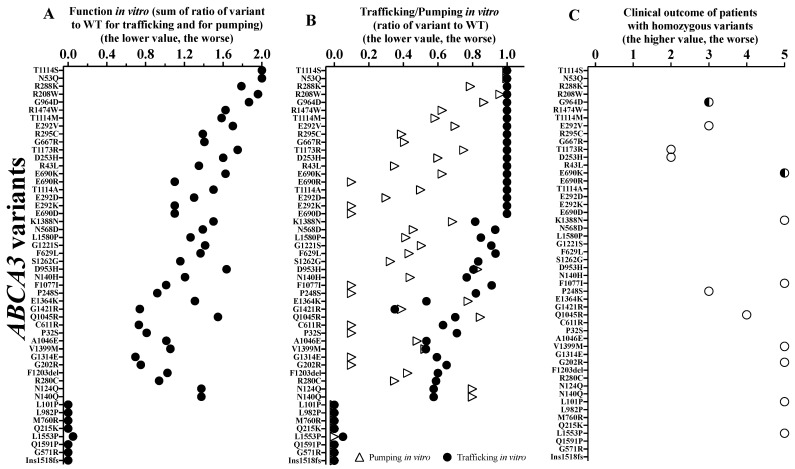
(**A**) Function in vitro; (**B**) trafficking/pumping of ABCA3 variants; (**C**) clinical outcome of patients with homozygous *ABCA3* variants. Hollow circles stand for the clinical outcomes of patients with homozygous variants. The semi-hollow circles stand for the clinical outcomes of patients with homozygous variants and with lung transplantation.

**Figure 6 ijms-24-07554-f006:**
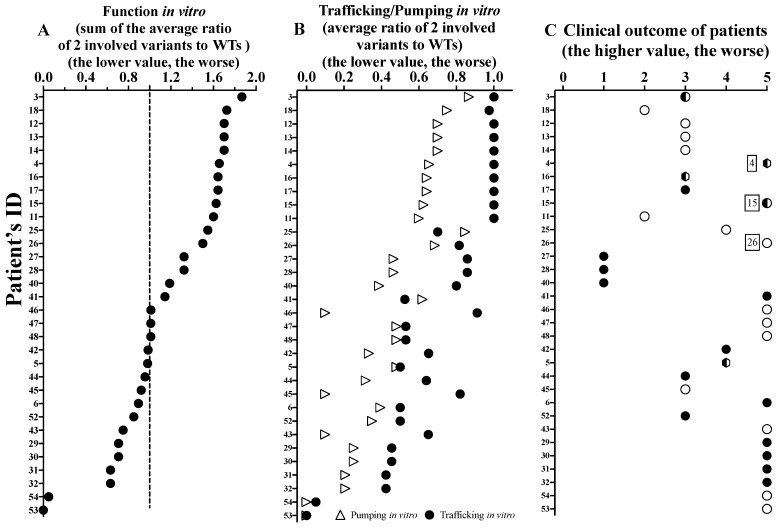
(**A**) Function in vitro of ABCA3 variants; (**B**) trafficking/pumping in vitro of ABCA3 variants; (**C**) clinical outcome of patients (32/54) with two variants that had been characterized in vitro, and sorting by sum function in vitro. The solid circles stand for the clinical outcomes of patients with compound heterozygous variants. The hollow circles stand for the clinical outcomes of patients with homozygous variants. The semi-hollow circles stand for the clinical outcomes of patients with homozygous variants and with lung transplantation. The semi-trapezoids stand for the clinical outcomes of patients with compound heterozygous variants and with lung transplantation. The function of the 2 variants (calculated as the sum of the average ratio of the 2 involved variants to the WTs) may correspond to the total functional activity of a subject with two alleles of *ABCA3*. The dotted line at 1 (=50%) stands for normal trafficking and pumping sum, i.e., a function of 2. The values without compatible results are marked with patients’ ID 4, 15, and 26, respectively (Appendix A).

**Table 1 ijms-24-07554-t001:** Reference values for the various assays obtained in WT ABCA3 transfected cells without treatment.

Code	Name of Assay	Absolute Results (Mean ± SD; n)	Mean Value Set to 1	Normalized SD (nSD; SD Expressed as a Fraction of the Mean Value Set to 1 in Each Experiment)		
				Mean	Median	Range
A	% ER localization of ABCA3	33.33 ± 23.63%; 1 **	1	0.71	0.71	-
B	N-glycosylation of ABCA3 *	63.12 ± 10.62%, 8	1	0.17	0.18	0.11–0.21
C	% Lysosomal localization of ABCA3	95.00 ± 8.66%; 1 **	1	0.09	0.09	-
D	Proteolytic cleavage of ABCA3	0.29 ± 0.10; 12	1	0.31	0.32	0.18–0.48
E	Volume of ABCA3^+^ vesicles	1.06 ± 0.17 µm^3^; 9	1	0.16	0.12	0.07–0.28
F	Amount of PC from recycling into ABCA3^+^ vesicles *	25,000 ± 4330 AU; 6	1	0.13	0.13	0.09–0.17
G	Amount of PC from de novo synthesis into ABCA3^+^ vesicles	65,149 ± 11,312 AU; 9	1	0.18	0.17	0.14–0.23
H	ATPase activity of ABCA3 *	100 ± 14.35%; 24	1	0.14	0.13	0.12–0.19

The mean values and standard deviation (SD) values were collected from experiments with wild-type cells in this study. * The mean values and corresponding SD values were collected as approximate values from graphs in the literatures due to the lack of exact values. If there was only an SEM approximate value, then the SD value was calculated manually as the SEM value multiplied by the square root of n (i.e., number of independent experiments). For further definition, the mean value of the SD was selected. ** Sixty randomly selected vesicles from three different fields.

## Data Availability

Original data can be made available upon request.

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
