# Peer review of "Quantifying Functional Impairment of ABCA3 Variants Associated with Interstitial Lung Disease"

_ijms, 2023, doi:10.3390/ijms24087554_

Round 1
Reviewer 1 Report
In this research, the authors characterized and quantified ABCA3 variants’ overall lipid transport function by assessing the in vitro impairment of its intracellular trafficking and pumping activity. Then, they expressed results relative to wild type, and generated data combined with previous results to correlate variants´ function and clinical phenotype. Acceptable quantitative variant-phenotype predictions were found and identified a threshold of about 50% loss of function to be associated with considerable morbidity and mortality due to ABCA3 dysfunction.
1. Why you didn't respect the classical order Introduction -> Materials and Methods-> Results-> Discussion-> Conclusion?
2. The supplement content is detailed and clear.
Reviewer 2 Report
The authors characterized and quantified ABCA3 variants’ overall lipid transport function by assessing the in vitro impairment of its intracellular trafficking and pumping activity.They have succesfully differentiated normal, impaired and defective variants.Yet, they found that more than 50% loss of function was correlated with morbidity and mortality . Prediction of genetic variants is crucial for the selection of the treatment .
It ia a scientifically sound paper,well written and presented.Their research protocol is well designed.The authors 8 different biochemical tests to characterize the ABCA3 variants that where statistically evaluated.Their results are nicely represented as well as, a figure showing the intracellular trafficking and function of wild type ABCA3 45 in alveolar type II cell is illustrated to provide a rapid synthetic approach of the subject.
Bibliography is up to date.
My suggestion is to ACCEPT and publish the paper in its present form.
